# Measurement Properties’ Evaluation of the Arabic Version of the Patient-Specific Functional Scale in Patients with Multiple Sclerosis

**DOI:** 10.3390/healthcare11111560

**Published:** 2023-05-26

**Authors:** Abdulrahman M. Alsubiheen, Nawaf A. Alzain, Alaa M. Albishi, Afaf A. M. Shaheen, Mishal M. Aldaihan, Muneera M. Almurdi, Abdulfattah S. Alqahtani, Asma A. Alderaa, Ali H. Alnahdi

**Affiliations:** 1Department of Health Rehabilitation Sciences, College of Applied Medical Sciences, King Saud University, P.O. Box 10219, Riyadh 11433, Saudi Arabia; 2Department of Rehabilitation, King Khalid University Hospital, King Saud University, Riyadh 11461, Saudi Arabia

**Keywords:** multiple sclerosis, disability, functional ability, outcome measures, reliability and validity

## Abstract

**Purpose**: The aim of this study was to assess the reliability and validity of the Arabic version of the patient-specific functional scale (PSFS-Ar) in patients with multiple sclerosis (MS) disorder. **Materials and Methods**: Reliability and validity were examined in patients with multiple sclerosis using a longitudinal cohort study design. One hundred (N = 100) patients with MS were recruited to examine the PSFS-Ar, test–retest reliability (using the interclass correlation coefficient model 2,1 (ICC_2,1_)), construct validity (using the hypothesis testing method), and floor–ceiling effect. **Results**: A total of 100 participants completed the PSFS-Ar (34% male, 66% female). The PSFS-Ar showed an excellent test–retest reliability score (ICC_2,1_ = 0.87; 95% confidence interval, 0.75–0.93). The SEM of the PSFS-Ar was 0.80, while the MDC_95_ was 1.87, indicating an acceptable measurement error. The construct validity of the PSFS-Ar was 100% correlated with the predefined hypotheses. As hypothesized, the correlation analysis revealed positive correlations between the PSFS-Ar and the RAND-36 domains of physical functioning (0.5), role limitations due to physical health problems (0.37), energy/fatigue (0.35), and emotional well-being (0.19). There was no floor or ceiling effect in this study. **Conclusions**: The study results showed that the PSFS-Ar is a self-reported outcome measure that is useful for detecting specific functional difficulties in patients with multiple sclerosis. Patients are able to express and report a variety of functional limitations easily and effectively, as well as to measure their response to physical therapy. The PSFS-Ar is, therefore, recommended for use in Arabic-speaking countries for clinical practice and research for patients with multiple sclerosis.

## 1. Introduction

Multiple sclerosis (MS) is an autoimmune disease that attacks the myelin sheaths around the axons of the central nervous system, leading to reduced mobility and adversely affecting the patient’s quality of life [1,2]. MS commonly affects young adults aged 20–40 years, with a higher incidence rate in women compared to men, with a ratio of 2:1 [1,2]. It is estimated that 2.8 million people worldwide suffer from MS [3]. The patient’s lifestyle and physical activity are restricted as the disease progresses, resulting in different levels of disability that eventually affects quality of life [4,5,6]. The health-related quality of life of people with MS is lower than that of people with general chronic diseases [7,8]. Patients with MS often experience fatigue, which is one of the main causes negatively affecting quality of life [9]. At some point in the course of MS, at least 75% of patients report fatigue as a symptom [10,11]. Most people consider fatigue as the most debilitating symptom, surpassing pain and disability, in terms of severity [12]. As a result of fatigue, jobs are lost and work hours are reduced, leading to substantial socioeconomic consequences [13]. MS can affect motor function, cognition, and sensation based on the location of damaged myelin, and eventually this can affect patient productivity [14]. It is not just the patient who suffers from MS but also the family and community who must cope with the disease [3]. It is important to note that the disease course varies extremely from patient to patient, and although there have been significant advancements in treatment, multiple sclerosis remains one of the leading causes of neurological disabilities in young people [15].

People with MS typically experience a range of disabilities, as described in the International Classification of Functioning, Disability, and Health (ICF) Core Set for MS [16,17]. According to the consensus among health practitioners, e.g., physical therapists and occupation therapists [17,18,19], it is agreed that patients with MS suffer from activity limitations and participation restrictions that increase the burden of disease and lower patients’ quality of life [20]. Therefore, reliable, rigorous, and valid measurements are needed to quantify and monitor activity limitations and disability in people with MS.

Patient-reported outcome measures (PROMs) fall under the category of outcome measurements, in which patients report changes in their health status without influence from healthcare providers [21,22,23,24,25,26,27]. The PROMs aim to assess changes in health status from the patient’s perspective and correlate strongly with changes in patient status [26,28,29]. One of the PROMs is the patient-specific functional scale (PSFS), which is a patient-centered outcome measure used to assess functional limitations in participants with functional disability during daily activities [23,24,25,27]. The PSFS was developed in 1995 by Stratford et al. [23], where patients report their perceived level of functional limitations and can be categorized as a patient-centered outcome measure as the patients report limitations in activities that they consider important and relevant to them [23,24,25,27]. The PSFS has no pre-specified list of items, but rather, each patient will list activities that matter to them; thus, the measure is tailored to the needs of each patient. It is the advantage of PSFS as a subjective scale to allow the patient to describe the most relevant activities that they have functional limitations with, which is absolutely different from patient to patient. Prior applications of the PSFS in various populations suggested that activities listed by patients will mostly fall under the ICF activity component [25,29]. The PSFS has been cross-culturally adapted to different languages and cultures, such as Brazilian, Swedish, Japanese, Nepali, Finnish, and Turkish languages [30,31,32,33,34,35,36].

The PSFS has been validated for use in many patient populations, such as individuals with musculoskeletal problems, spine conditions, and Parkinson’s disease [37]. Recent studies have explored the feasibility and informativeness of the PSFS for identifying activities that persons with Parkinson’s disease self-identify as difficult [29]. However, evaluating the measurement properties of the PSFS for patients with MS remains largely unexplored. The PSFS was translated to the Arabic language and validated in patients with musculoskeletal disorders [28].

Patients with MS usually suffer from functional limitations [38]. The PSFS is a subjective measurement reported by the patient regarding a relevant functional limitation. No prior studies have examined the psychometric properties of this important scale (PSFS-Ar) in patients with MS. Therefore, this study aims to examine the measurement properties of the Arabic version of the PSFS in patients with MS.

## 2. Materials and Methods

### 2.1. Study Design

In the study of patients with MS, the reliability and validity of the PSFS-Ar were examined using a longitudinal cohort study method. The study was ethically approved by the Institutional Review Board at King Saud University (no. E-19-4530). Informed consent was obtained from all the participants involved in the study.

### 2.2. Settings and Participants

Participants in this study were recruited if they had a confirmed diagnosis of MS from a neurologist at King Khalid University Hospital (KKUH). Patients with MS were invited to participate in the study. All patients with MS signed the consent form and completed the study without recording any withdrawals. Patients with MS were included in the study, regardless of the stage of the disease. The study included participants who were 18 years of age or older and who were capable of communicating, understanding the questions, walking with or without assistance, and reading and writing in Arabic. The study excluded participants with other neurological diseases and relapses less than 30 days prior to assessment. The study was conducted at King Khalid University Hospital’s (KKUH) Physical Therapy Outpatient Clinic in Riyadh, Saudi Arabia.

### 2.3. Outcome Measures

#### 2.3.1. The Patient-Specific Functional Scale (PSFS)

A total of three to five activities were listed as difficult or impossible by participants [23]. Each participant was requested to rate each of the listed activities on a scale of 11 points. In the scale anchors, activities ranged from 0 (could not be completed) to 10 (could be completed at the same level as before the disease). Items scores were averaged to produce the total PSFS score. A higher PSFS total score indicates better functional ability. In this study, the Arabic version of the PSFS (PSFS-Ar) was used. Alnahdi et al. translated the PSFS-Ar into Arabic for use with patients suffering from lower-extremity orthopedic conditions, and the measure has been found to be valid and reliable [28]. The PSFS has no fixed item and no subscale. Each patient reported 3–5 tasks that he/she felt were difficult. For example, patient one reported walking a short distance, sitting to standing, and turning in bed, and patient two reported running, showering, and gardening. The scores of the 3 to 5 items listed by the participants were averaged to provide a total score (0–10), representing the latent variable that is activity limitation.

#### 2.3.2. RAND 36-Item Health Survey (RAND-36)

The RAND-36 assesses general health, addressing concepts relevant to people of all ages, diseases, and treatments. [39]. It is a self-administered PROM, with eight subscales: physical functioning, role limitations due to physical health problems, pain, general health, social functioning, role limitations due to emotional problems, energy/fatigue, and emotional well-being. Each subscale is scored on a 0–100 scale, with 0 representing the worst possible health status and 100 representing the best possible health status [39]. The RAND-36 is a valid and reliable measure for patients with MS [39]. In this study, an Arabic version of the RAND-36 was used with evidence supporting its validity and reliability. [40].

#### 2.3.3. Global Rating of Change (GRC)

The Global Rating of Change (GRC) scale is a commonly used outcome measure to monitor changes in patients’ health status [41]. A GRC scale from −5 to 5 was used in the current study. The range of scores from 1 to 5 indicates an improvement in health status compared to the first session, whereas scores from −1 to −5 indicate deterioration, with zero indicating no change in health status. Patients with scores ranging from −1 to 1 in the GRC were considered to have stable health status between the two administrations of the PSFS-Ar [41].

### 2.4. Procedure

During the first session, participants completed the RAND 36-item Health Survey and PSFS-Ar for the first time (T1). In the second session, which was 4 to 7 days after the first session, participants completed the PSFS-Ar for the second time (T2) to assess the test–retest reliability of the PSFS-Ar. In the second session, participants were also asked to rate the change in their health status using the Global Rating of Change (GRC) scale to determine that the patient’s health status did not change between T1 and T2. Participants signed inform consent forms prior to participation.

## 3. Statistical Analysis

### 3.1. Test–Retest Reliability

Test–retest reliability for the PSFS-Ar was examined among participants who completed the PSFS-Ar twice and reported no change in their condition between the two testing sessions. Test–retest reliability was examined using the Intraclass Correlation Coefficient model 2,1 (ICC_2,1_), with a 95% confidence interval for absolute agreement, whereby an ICC value equal to or greater than 0.7 was considered to indicate sufficient reliability [42]. To ensure that the participants’ conditions did not change between the test and retest sessions, participants were required to complete the GRC in the retest session. For the GRC, patients reporting 1 (tiny bit worse, almost the same), 0 (no change), or 1 (tiny bit worse, almost the same) were considered unchanged and were included in the test–retest reliability analysis.

### 3.2. Measurement Error

To examine the measurement error associated with repeated measurements using the PSFS-Ar, the standard error of measurement (SEM) and the Bland–Altman plot were used. The SEM was calculated using the formula SEM = SD × √(1 − ICC), where SD is the pooled standard deviation and ICC is the test–retest intraclass correlation coefficient [43]. The minimal detectable change with 95% confidence (MDC_95_) was used to quantify the true change in the PSFS-Ar beyond the measurement error. The MDC_95_ was calculated using the following formula: MDC_95_ = SEM × 1.96 × √2 [43].

### 3.3. Floor and Ceiling Effects

Floor and ceiling effects of the PSFS-Ar were considered present if more than 15% of the participants reached the lowest score (0) or the highest score (10). If the floor or ceiling effects are present, this leads to limited content validity [42].

### 3.4. Construct Validity

The construct validity of the PSFS-Ar as a measure of activity and participation in patients with MS was assessed by examining the following four pre-defined hypotheses: (1) the PSFS-Ar was hypothesized to have at least a moderate positive correlation with the physical functioning domain of the RAND-36; (2) the PSFS-Ar was hypothesized to have at least a moderate positive correlation with the role limitations due to the physical health problems domain of the RAND-36; (3) the PSFS-Ar was hypothesized to have at least a moderate positive correlation with the energy/fatigue domain of the RAND-36; (4) the PSFS-Ar was hypothesized to have a weak correlation with the emotional well-being domain of the RAND-36.

The strength of the correlation was considered moderate if the absolute value of the correlation coefficient ranged from ≥0.3 to <0.6 and a weak correlation if it was <0.3, with ≥0.60 indicating a good correlation [44]. For normally distributed data, Pearson correlation (r) was used to examine the hypothesized correlations, and the Spearman correlation (RS) was used for non-normally distributed data. The construct validity of the PSFS-Ar was considered sufficient if at least 75% of the hypotheses were supported [42]. All statistical analyses were conducted using SPSS version 25.0 (IBM Corp., Armonk, NY, USA). The level of significance was set at 0.05.

### 3.5. Sample Size Estimation

The minimum required sample size was based on the consensus-based standards for the selection of health measurement instruments (COSMIN) that recommended a sample size of 50 as the minimum requirement to investigate floor and ceiling effects and construct validity [45]. For test–retest reliability, it was determined that 30 participants would be the minimum sample size required [45].

## 4. Results

A total of 100 patients with MS participated in the current study. Their demographic characteristics are presented in Table 1. The descriptive statistics of the RAND-36 and the PSFS-Ar are provided in Table 2. Participants had no missing items in the PSFS-Ar and RAND-36.

### 4.1. Test–Retest Reliability

For the test–retest reliability, 50 participants completed the PSFS-Ar twice. Thirty-three participants were included in the test–retest reliability assessment given that they had GRC scores between −1 and 1, which indicates no change in their health condition. Seventeen participants were excluded from test–retest reliability assessment given that they reported a change in their health condition between the two testing sessions (according to their GRC scores). The relative reliability of the PSFS-Ar was excellent (ICC_2,1_ = 0.87; 95% confidence interval 0.75–0.93). The mean score and standard deviation values for the PSFS-Ar were 5.43 ± 2.19 and 5.33 ± 2.10 for the first and second testing sessions, respectively (Table 3).

### 4.2. Measurement Error

On the basis of the ICC reliability coefficient, the SEM of the PSFS-Ar was 0.80, while the MDC_95_ was 1.87 (Table 3). As shown in the Bland–Altman plot, there was no systematic bias along the different levels of the scale but rather a random error (Figure 1). The 95% CI of the mean difference between the second testing session and the first testing session lies on the line of equality (zero), indicating that there was no systematic error between the testing sessions.

### 4.3. Construct Validity

In order to assess the construct validity of the PSFS-Ar, the pre-defined hypotheses regarding the correlation between the PSFS-Ar and the different domains of the RAND-36 were tested. Non-parametric Spearman’s correlation was used because the data were not normally distributed. As hypothesized, the PSFS-Ar showed moderate positive correlation with the physical functioning domain of the RAND-36, moderate positive correlations with the role limitations due to physical health problems domain of the RAND-36, and moderate positive correlations with the energy/fatigue domain of the RAND-36. The PSFS-Ar also showed weak correlation with the emotional well-being domain of the RAND-36. The correlations between the PSFS-Ar and different RAND-36 domains are shown in Table 4.

### 4.4. Floor and Ceiling Effects

Three patients with MS who participated in this study achieved the highest score of 10 (3%), and no participants achieved the lowest score of 0, representing 0% of the total sample. Thus, the PSFS-Ar did not have any floor or ceiling issues.

## 5. Discussion

The purpose of this study was to evaluate the measurement properties of the PSFS-Ar for patients with MS, including test–retest reliability, measurement error, construct validity, and floor and ceiling effects. The results of the current study support the test–retest reliability and construct validity of the PSFS-Ar and suggest that the scale has acceptable measurement error with no floor or ceiling issues.

The test–retest reliability of the PSFS-Ar was evaluated using the ICC model 2,1. An ICC greater than 0.70 is considered to have acceptable retest reliability according to current recommendations [42]. In the current study, the GRC scale was used to ensure that participants included in the test–retest reliability had unchanged health status between the two testing sessions. Based on the test–retest reliability findings of this study, the PSFS-Ar appears to be an appropriate reliable outcome measure. The point estimate of the ICC for the PSFS-Ar in Table 3 and also the lower limit of the 95% confidence interval were greater than 0.7, indicating acceptable reliability [42,46]. There was a similar PSFS-Ar reliability estimate in this study in patients with MS (ICC = 0.87) compared with those in patients with orthopedic problems (ICC = 0.86) [28]. Based on a systematic review of 22 studies, Pathak et al. determined that the PSFS in orthopedic conditions had a test–retest reliability coefficient (ICC) between 0.71 and 0.98, and the ICC was also reported to be 0.72 in patients with Parkinson’s disease [37].

For the purpose of quantifying the PSFS-Ar measurement error, the SEM and MDC were used (Table 3). In order to describe a change in the functional status of a patient based on the MDC, the Arabic PSFS must change by at least 1.87. Based on the 18.7% MDC of the PSFS-Ar, the magnitude of the MDC seems acceptable and appropriate for clinical use. Compared to previous studies, the MDC reported in this study for the PSFS-Ar was consistent with the literature’s range values. In a systematic review, the MDC value for the PSFS in patients with orthopedic conditions was reported to range from 0.64 to 3.3 [37].

The correlational findings in this study support the construct validity of the PSFS-Ar as a measure of activity limitation in patients with MS, as 100% of the pre-defined construct validity assumptions were supported [42]. Given the fact that both RAND-36 physical functioning and the PSFS-Ar are measures of activity limitation, it was predicted that they would correlate. The findings of the present investigation confirmed this concept, which was demonstrated in a previous study [28], as the PSFS-Ar has been reported to be associated with RAND-36 physical functioning in individuals with orthopedic conditions (*r* = 0.64) in comparison to (*r* = 0.50) in this investigation. In a systemic review, the PSFS and RAND-36 physical functioning were correlated between *r* = 0.22 and *r* = 0.83 in lower-extremity conditions [37]. Moreover, among other MS scales that measure physical functioning, Hobrat et al. used the 12-Item MS walking scale (MSWS), which measures walking ability as a physical function. In the study, they found a strong correlation between the MSWS and the physical functioning domain of SF-36 (*r* = −0.79) [47]. Hobrat et al. demonstrated a strong correlation between the MSWS and the SF-36 physical functioning domain in patients with MS, and our study supported the strong correlation between the PSFS and RAND-36 physical function. Therefore, the PSFS can be used to measure physical activity in patients with MS.

As expected, this study showed that there is a weak correlation between the PSFS-Ar and the emotional well-being domain of the RAND-36 (*r* = 0.19). The PSFS-Ar and the emotional well-being domain of the RAND-36 measure different constructs and, thus, were expected to show a low correlation. The low correlation reported in our study is in line with a previous study (*r* = 0.13) [28]. The correlation of the PSFS-Ar with the role limitations due to the physical health problem domain of the RAND-36 was moderate (*r* = 0.37). This correlation was expected, as physical health problems could affect, and be related to, the PSFS score that reflects a similar construct (activity limitation). Furthermore, the correlation of the PSFS-Ar with the energy/fatigue domain of the RAND-36 was moderate (*r* = 0.35). Fatigue is one of the most common symptoms of MS [48]. The results indicate how fatigue affects the PSFS score and how it is linked to fatigue. The correlation values will help the practitioners to measure the functional limitations and track treatment improvements, which eventually help to improve the healthcare system.

There were no floor or ceiling effects in the PSFS-Ar, where only 3% of participants achieved the highest score and 0% achieved the lowest score. In the presence of a floor and ceiling effect, 15% of participants will achieve the highest or lowest score [42]. In the current study, no floor or ceiling effect was observed, suggesting that the PSFS-Ar may be suitable for identifying specific difficulties of functional activities among patients with MS. The results of this study were consistent with previous reports [27,28].

There are many practical implications of the PSFS-Ar. It can be used in daily clinical practice as well as in research studies to measure activity limitations in Arabic-speaking patients with MS. The PSFS-Ar allows rehabilitation specialists to quantify activity limitation, according to the culture and lifestyle of Arabic speakers with MS. Moreover, rehabilitation specialists can confidently interpret patient’s scores in the PSFS-Ar to represent the extent of activity limitation.

In this study, there are some limitations. Participants were not objectively assessed on their cognitive abilities. During interaction with the participants, researchers subjectively checked that all participants were cognitively capable of completing the outcome measures. Moreover, the responsiveness of the PSFS-Ar in patients with MS has not been examined. Thus, research on the responsiveness of the PSFS-Ar to patients with MS is needed in the future.

## 6. Conclusions

To the best of our knowledge, this is the first study to assess PSFS-Ar measurement properties in patients with MS. The study results show that the PSFS-Ar is a self-reported outcome measure that is useful for detecting specific functional difficulties in patients with multiple sclerosis. Patients are able to express and report a variety of functional limitations easily and effectively, as well as measure their response to physical therapy. The PSFS-Ar has excellent test–retest reliability, acceptable measurement errors, and evidence supporting its construct validity with no floor or ceiling problems. The PSFS-Ar is, therefore, recommended for use in Arabic-speaking countries for clinical practice and research for patients with MS.

## Figures and Tables

**Figure 1 healthcare-11-01560-f001:**
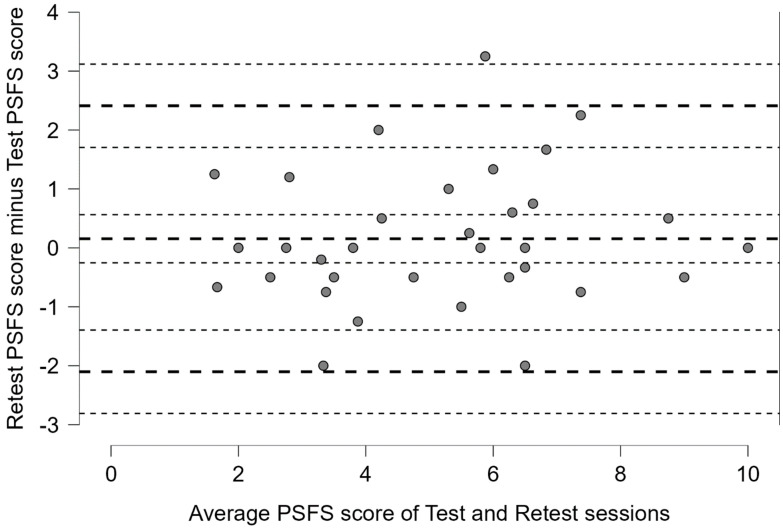
Bland–Altman plot of the difference (vertical axis) versus the mean (horizontal axis) for measurements obtained in the first testing session (test) and in the second testing session (retest). The middle bold-dashed line represents the mean difference (with 95% confidence interval) between the two testing sessions. The upper and lower bold-dashed lines represent the 95% limits of agreement with a 95% confidence interval for the upper and lower limits.

**Table 1 healthcare-11-01560-t001:** Participant characteristics (N = 100).

Variable	Mean ± SD or N (%)
Age (years)	33.73 ± 8.87
Gender	Male = 34 (34%)
	Female = 66 (66%)
Handedness	Right = 61 (61%)
	Left = 15 (15%)
	Ambidextrous = 24 (24%)
Height (cm)	163.2 ± 9.24
Weight (kg)	73.65 ± 24.30
Body Mass Index (kg/m^2^)	27.51 ± 8.35
Family history with MS	Yes: 16 (16%)
	No: 84 (84%)
Marital status	Single: 45 (45%)
	Married 45 (45%)
	Divorced 10 (10%)
Level of education	Elementary school: 2 (2%)
	Middle school: 14 (14%)
	High-school: 14 (14%)
	Bachelor degree: 68 (68%)
	Post-graduate study: 2 (2%)
Onset of MS (months) **	62.00 (95.00) *

* Median (interquartile range). ** N = 83.

**Table 2 healthcare-11-01560-t002:** Outcome measures for all participants (N = 100).

Variable	Mean ± SD
PSFS-Ar	5.43 ± 2.19
RAND-36 physical functioning	69.35 ± 27.17
RAND-36 role limitation due physical health problems	51.25 ± 43.42
RAND-36 energy/fatigue	50.40 ± 25.88
RAND-36 emotional well-being	60.92 ± 24.16

**Table 3 healthcare-11-01560-t003:** PSFS-Ar test–retest reliability and measurement error (N = 33).

	Mean ± SD	Mean Difference ^a^ (95% CI)	ICC_2,1_ (95% CI)	SEM	MDC_95_
**Test**	5.43 ± 2.19	0.155 (−0.25 to 0.56)	0.87 (0.75–0.93)	0.80	1.87
**Retest**	5.33 ± 2.10				

ICC: intraclass correlation coefficient (two-way random model for agreement); SEM: standard error of measurement for agreement; MDC_95_: minimal detectable change with a 95% confidence. ^a^ Test score minus the retest score.

**Table 4 healthcare-11-01560-t004:** Correlation between the PSFS-Ar and different RAND-36 domains (N = 100).

Variable	rho (95% CI)	*p*
RAND-36 physical functioning	0.50 (0.33 to 0.64)	<0.001
RAND-36 role limitations due to physical health problems	0.37 (0.19 to 0.53)	<0.001
RAND-36 energy/fatigue	0.35 (0.14 to 0.52)	<0.001
RAND-36 emotional well-being	0.19 (−0.13 to 0.39)	0.053

CI = confidence interval; *p* = *p* value; rho = Spearman’s correlation.

## Data Availability

The datasets generated during the current study are available from the corresponding author upon reasonable request.

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
