# Peer review of "Measurement Properties’ Evaluation of the Arabic Version of the Patient-Specific Functional Scale in Patients with Multiple Sclerosis"

_healthcare, 2023, doi:10.3390/healthcare11111560_

Round 1
Reviewer 1 Report
The correlation value of physical functioning is 0.5, role limitations due to physical health problems is 0.37, energy/fatigue is 0.35, and emotional well-being is 0.19. From this obtained value, what is the suggestion or recommendation to improve the healthcare system for these disabled patients?
Author Response
The correlation value of physical functioning is 0.5, role limitations due to physical health problems is 0.37, energy/fatigue is 0.35, and emotional well-being is 0.19. From this obtained value, what is the suggestion or recommendation to improve the healthcare system for these disabled patients?
Thank you for the comment. We added this statement in lines 301 -303
The correlation values will help the practitioners to measure the functional limitations and track treatment improvements which eventually help to improve the healthcare system.

Reviewer 2 Report
The authors have done interesting work that aims to assess the reliability and validity of the Arabic version of the patient-specific functional scale in patients with multiple sclerosis.
The introduction reflects the problem to be dealt with and correctly explains the background and objective of said work.
The methods section is one of the most important in an article and must be modified for publication:
- A first subsection should be added that specifies the study design and the data on the Ethics Committee. The second subsection would be that of participants (which is already included).
- In the subsection of participants, a flow diagram of the participants should be added that reflects the possible losses and the total sample.
- To give a higher quality to the study, the "Procedure" section should be placed after the "outcome measures" section.
The results section is well structured and complete. In the discussion section, Table 5 appears. Within this section, the use of these resources is not correct, so if said information is very relevant, it should appear in the results section. In the discussion section, the results found with other previous studies should be discussed.
The references section must be modified. The references do not meet the journal's standards.
Author Response
(Reviewer 2)
The authors have done interesting work that aims to assess the reliability and validity of the Arabic version of the patient-specific functional scale in patients with multiple sclerosis.
Thank you
The introduction reflects the problem to be dealt with and correctly explains the background and objective of said work.
Thank you
The methods section is one of the most important in an article and must be modified for publication:
- A first subsection should be added that specifies the study design and the data on the Ethics Committee. The second subsection would be that of participants (which is already included).
Thank you for the comment. We added subsection for the study design (line 78)
2.1. Study design
In the study of patients with MS, the reliability and validity of the PSFS-Ar were examined using a longitudinal cohort study method. The study was ethically approved by the Institutional Review Board at King Saud University (no. E-19-4530). Informed consent was obtained from all the participants involved in the study.
- In the subsection of participants, a flow diagram of the participants should be added that reflects the possible losses and the total sample.
Thank you for the comment. In this study, there are no dropouts, and the section "sample size estimation" has included the total sample.
- To give a higher quality to the study, the "Procedure" section should be placed after the "outcome measures" section.
Thank you for the comment. We placed the "Procedure" section after the "outcome measures" section.
The results section is well structured and complete. In the discussion section, Table 5 appears. Within this section, the use of these resources is not correct, so if said information is very relevant, it should appear in the results section. In the discussion section, the results found with other previous studies should be discussed.
Thank you for the comment. We agreed to delate table 5 as the pre-defined hypotheses have been mentioned under “Statistical Analysis” in the “Construct Validity” section
The references section must be modified. The references do not meet the journal's standards.
Thank you for the comment. We updated the citation using MDPI.ens references style for EndNote as instructed in the journal guidelines

Reviewer 3 Report
- English writing needs substantial revisions.
- citation should be according to the journal guidelines.
- lines 54-55: validated for what kind of population? describe in detail.
- more information regarding the validity and reliability of the PSFS should be described. In which countries has it been validated? with which diseases? how many items and what kind of scale is used to rate their perception? this should be discussed in the introduction.
- Thus, the overall introduction failed to convince me that this study is needed in the literature since no limitations and gaps in previous research are presented. in fact, the authors do not describe the population with MS nor the implications of the scale.
Methods
- Participant recruitment procedures should be described in full. Who did the diagnosis? how many clinics (e.g., hospitals, clinics) were contacted? How many did not consent to participate? This and more details should be reported for transparency. The type of data collection sampling should be described as well as the study design.
- Item examples should be reported for each scale and subscale.
- How were latent variables transformed into observed ones? please clarify.
- At this point, validity is referred to the extent to which the items on a scale adequately cover the full range of the construct being measured. The authors conducted only construct validity, which limits the interpretation of measurement validity. More analyses are needed to regard the PSFS such as factor analyses.
Results
- Table 1: is there any contribution to the objective of the study to measure and report individual variables (e.g., marital status, level of education) related to MS?
- Why is the Onset of MS reported in the median and not the mean?
Discussion
- limitations are not reported.
- the practical implications should be explored in full.
- table 5 is odd since no hypotheses are reported in the introduction section.
Author Response
(Reviewer 3)
Comments and Suggestions for Authors
- English writing needs substantial revisions.
Thank you for your comment. The manuscript undergone for English language editing by MDPI. I have received an English Editing Certificate
- citation should be according to the journal guidelines.
Thank you for the comment. We updated the citation using MDPI.ens references style for EndNote as instructed in the journal guidelines
- lines 54-55: validated for what kind of population? describe in detail.
Thank you for the comment. We added the other population (lines 64- 65)
The PSFS has been validated for use in many patient populations, such as individuals with musculoskeletal problems, spine conditions, and Parkinson’s disease [27].
- more information regarding the validity and reliability of the PSFS should be described. In which countries has it been validated? with which diseases? how many items and what kind of scale is used to rate their perception? this should be discussed in the introduction.
Regarding this point, we added and modified in the introduction:
One of the PROMs is the patient-specific functional scale (PSFS) which is a patient-centered outcome measure used to assess functional limitations in participants with a functional disability during daily activities [13-15,17]. The PSFS was developed in 1995 by Stratford et al. [13] where patients report their perceived level of functional limitations and can be categorized as a patient-centered outcome measures as the patients report limitations in activities that they consider important and relevant to them [13-15,17]. The PSFS has no pre specified list of items, but rather each patient will list activities that matter to them, thus the measure is tailored to the needs of each patient. Prior applications of the PSFS in various population suggested that activity listed by patients will mostly fall under the ICF activity component [15,19]. The PSFS has been cross-culturally adapted to different languages and cultures, such Brazilian, Swedish, Japanese, Nepali, Finnish, and Turkish languages [20-26].
The PSFS has been validated for use in many patient populations, such as individuals with musculoskeletal problems, spine conditions, and Parkinson’s disease [27]. Re-cent studies have explored the feasibility and informativeness of the PSFS for identifying activities that persons with Parkinson's disease self-identify as difficult [19]. However, evaluating the measurement properties of the PSFS for patients with MS remains largely unexplored. The PSFS was translated to the Arabic language and validated in patients with musculoskeletal disorders [18].
Also in the section of outcomes measures, we added, updated and this point
2.3. Outcome measures
2.3.1. The Patient-Specific Functional Scale (PSFS)
A total of three to five activities were listed as difficult or impossible by participants [29]. Each participant was requested to rate each of the listed activity on a scale of 11 points. In the scale anchors, activities ranged from 0 (could not be completed) to 10 (could be completed at the same level as before the disease). Items scores were averaged to produce the total PSFS score. A higher PSFS total score indicates better functional ability. In this study, the Arabic version of the PSFS (PSFS-Ar) was used. Alnahdi et al. translated the PSFS-Ar into Arabic for use with patients suffering from lower-extremity orthopedic conditions, and the measure has been found to be valid and reliable [18]. The PSFS has no fixed item and no subscale. Each patient reported 3-5 tasks that he/she felt difficult. For example, patient one reported: walking a short distance, sitting to standing, and turning in bed, and patient two reported: running, showering, and gardening. The scores of the 3 to 5 items listed by the participants were averaged to provide a total score (0-10), representing the latent variable that is activity limitation.
- Thus, the overall introduction failed to convince me that this study is needed in the literature since no limitations and gaps in previous research are presented. in fact, the authors do not describe the population with MS nor the implications of the scale.
Thank you for the comment. We added (line 72 -75)
Patients with MS usually suffer from functional limitations [28]. The PSFS is a subjective measurement reported by the patient regarding a relevant functional limitation. No prior studies have examined psychometric properties of this important scale (PSFS-Ar) in patients with MS. Therefore, this study aims to examine the measurement properties of the Arabic version of the PSFS in patients with MS.
Methods
- Participant recruitment procedures should be described in full. Who did the diagnosis? how many clinics (e.g., hospitals, clinics) were contacted? How many did not consent to participate? This and more details should be reported for transparency. The type of data collection sampling should be described as well as the study design.
Thank you for the comment. We added (lines 85-88):
Participants in this study were recruited if they had a confirmed diagnosis of MS by a neurologist in King Khalid University Hospital (KKUH). Patients with MS were invited to participate in the study. All patients with MS signed the consent form and completed the study without recording any withdrawals.
- Item examples should be reported for each scale and subscale.
Thank you for the comment. We added (lines104-107):
The PSFS has no fixed item and no subscale. Each patient reported 3-5 tasks that he/she felt difficult. For example, patient one reported: walking a short distance, sitting to standing, and turning in bed, and patient two reported: running, showering, and gardening.
- How were latent variables transformed into observed ones? please clarify.
Thank you for the comment. We added (lines 107-109):
The scores of the 3 to 5 items listed by the participants were averaged to provide a total score (0-10), representing the latent variable that is activity limitation.
- At this point, validity is referred to the extent to which the items on a scale adequately cover the full range of the construct being measured. The authors conducted only construct validity, which limits the interpretation of measurement validity. More analyses are needed to regard the PSFS such as factor analyses.
We established the construct validity for the Arabic PSFS using a method recommended by the COSMIN guideline, that is hypothesis testing of specific pre-defined hypotheses. Factor analysis is not applicable for the PSFS given that items within the PSFS are not fixed but different for each individual patient.
Results
- Table 1: is there any contribution to the objective of the study to measure and report individual variables (e.g., marital status, level of education) related to MS?
To give reader a background information about the participants.
- Why is the Onset of MS reported in the median and not the mean?
The data was not normally distributed.
Discussion
- limitations are not reported.
Thank you for the comment. We added (lines 318-323):
In this study, there are some limitations. Participants were not objectively assessed on their cognitive abilities. During interaction with the participants, researchers subjectively checked that all participants were cognitively capable of completing the outcome measures. Moreover, the responsiveness of the PSFS-Ar in patients with MS has not been examined. Thus, research on the responsiveness of the PSFS-Ar to patients with MS is needed in the future.
- the practical implications should be explored in full.
Thank you for the comment. We added (Lines 311-316):
There are many practical implications of the PSFS-Ar. It can be used in daily clinical practice as well as in research studies to measure activity limitations in Arabic-speaking patients with MS. The PSFS-Ar allows rehabilitation specialists to quantify activity limitation, according to the cultural and lifestyle of Arabic speakers with MS. Moreover, rehabilitation specialists can confidently interpret patient’s scores in the PSFS-Ar to represent the extent of activity limitation.
- table 5 is odd since no hypotheses are reported in the introduction section.
Thank you for the comment. We agreed to delete table 5 as the pre-defined hypotheses have been mentioned under “Statistical Analysis” in the “Construct Validity” section

Reviewer 4 Report
Dear Authors,
Thanks for your cooperation in conducting this paper.
The article entitled “Measurement Properties Evaluation of the Arabic Version of the Patient-Specific Functional Scale in Patients with Multiple Sclerosis” seems well for publication in my opinion.
The title is interesting and the context of the study was informative and useful as well, could be widely used in Arab nations for patients with MS to investigate the quality of life better in long-term follow-up. The main concern in translating questionnaires is to interpret them in simple and understandable languages for patients. It would be better if you attach the Arabic questionnaire form to publish with the paper. The method and results are well-written and discussed as well. The conclusion reflected the results well.
Author Response
(Reviewer 4)
Comments and Suggestions for Authors
Dear Authors,
Thanks for your cooperation in conducting this paper.
The article entitled “Measurement Properties Evaluation of the Arabic Version of the Patient-Specific Functional Scale in Patients with Multiple Sclerosis” seems well for publication in my opinion.
Thank you
The title is interesting and the context of the study was informative and useful as well, could be widely used in Arab nations for patients with MS to investigate the quality of life better in long-term follow-up. The main concern in translating questionnaires is to interpret them in simple and understandable languages for patients. It would be better if you attach the Arabic questionnaire form to publish with the paper. The method and results are well-written and discussed as well. The conclusion reflected the results well.
Thank you for your valuable comments, the translated version of PSFS was published in another publication. Here is the link:
Alnahdi AH, Murtada BA, Zawawi AT, Omar MT, Alsobayel HI. Cross-cultural adaptation and measurement properties of the Arabic version of the Patient-Specific Functional Scale in patients with lower extremity musculoskeletal disorders. Disabil Rehabil. 2022 Jul;44(15):4104-4110. doi: 10.1080/09638288.2021.1880651. Epub 2021 Feb 15. PMID: 33587649.
Round 2
Reviewer 3 Report
The authors did a good job reviewing their manuscript.
Minor spell check is required.
Author Response
Your comments are greatly appreciated. For best quality of proofreading, we sent this manuscript to English Editing Department at MDPI. The MDPI confirmation certificate can be found after the references at the end of the manuscript. Moreover, for final review, we sent this manuscript again to another native English-speaking colleague who made minor changes as indicated.
